

# Weekly to monthly terminus variability of Greenland's marine-terminating outlet glaciers

Taryn E. Black[1,2], Ian Joughin[2]

[1]Department of Earth and Space Sciences, University of Washington, Seattle, WA, 98195, USA
[2]Polar Science Center, Applied Physics Laboratory, Seattle, WA, 98105, USA

*Correspondence to*: Taryn E. Black (black.taryn.e@gmail.com)

**Abstract.** Earlier studies have shown that seasonal terminus-position variability is superimposed on multi-decadal trends of glacier retreat. To characterize this seasonal variability, we manually digitized terminus positions for 219 marine-terminating glaciers in Greenland from January 2015 through December 2021 using Sentinel-1 SAR mosaics. We digitized at a monthly
frequency for 199 glaciers and at a six-day frequency for 20 glaciers. We found that nearly 75% of glacier termini in Greenland vary significantly on a seasonal basis. For these seasonally-varying glaciers, on average, seasonal retreat typically begins in mid-May, and seasonal advance generally commences in early October. The timing of the initiation of the retreat period may be related to the timing of the onset of ice-sheet surface melt. The rate of retreat events peaks in late summer and reaches a minimum in late winter and early spring. The median magnitude of terminus-position seasonality, the difference
between glacier length at the dates of peak advance and retreat, is about 220 m. We find a stronger correlation between this magnitude and glacier velocity than between magnitude and glacier width. Terminus-position seasonality can influence longer-term glacier dynamics and, consequently, ice-sheet mass balance. This study contributes to our understanding of terminus-position seasonality for individual glaciers and collectively for glaciers around the entire Greenland Ice Sheet.

## 1 Introduction

The majority of marine-terminating outlet glaciers in Greenland have retreated over the past several decades, and regionally this retreat accelerated in the 1990s and 2000s (Black and Joughin, 2022; Carr et al., 2017; Fahrner et al., 2021; Howat and Eddy, 2011; King et al., 2020). This terminus position retreat is linked to increasing ice discharge (King et al., 2020, 2018; Mouginot et al., 2019). Between 2012 and 2017, ice discharge contributed to ~43% of the net mass loss from the Greenland Ice Sheet (Shepherd et al., 2020), and projections of future ice-sheet mass loss indicate that discharge will continue to be an
important contributor through 2100 (Choi et al., 2021).

The multidecadal behavior of the terminus positions of marine-terminating outlet glaciers in Greenland is well-characterized (Black and Joughin, 2022; Fahrner et al., 2021; Goliber et al., 2022; Howat and Eddy, 2011; King et al., 2020). Superimposed on these multidecadal trends, many glaciers exhibit seasonal terminus-position variability, which is typically expressed as wintertime advance and summertime retreat (Carr et al., 2013; Cassotto et al., 2015; Fried et al., 2018; Howat et
al., 2010; Joughin et al., 2008b; Kehrl et al., 2017; Kneib-Walter et al., 2021; Moon et al., 2015; Murray et al., 2015; Sakakibara and Sugiyama, 2019; Schild and Hamilton, 2013; Seale et al., 2011). Seasonal terminus-position variability





spatially varies in amplitude (Fried et al., 2018; Howat et al., 2010; Kehrl et al., 2017; Moon et al., 2015; Seale et al., 2011), and it has been suggested that the amplitude may depend on glacier width (Schild and Hamilton, 2013; Seale et al., 2011) or calving style (Fried et al., 2018). Most studies of seasonal terminus-position variability have examined either Greenland's

largest outlet glaciers (Cassotto et al., 2015; Joughin et al., 2008b; Kehrl et al., 2017; Schild and Hamilton, 2013), a regional subset of glaciers (Carr et al., 2013; Fried et al., 2018; Howat et al., 2010; Moon et al., 2015; Sakakibara and Sugiyama, 2019; Seale et al., 2011), or a small number of glaciers around the ice sheet (Bevan et al., 2012).

Previous studies have suggested that seasonal terminus-position variability is driven by the effect of proglacial mélange or meltwater runoff on calving rates. In front of some glaciers, a rigid mélange tends to form in the winter as sea ice freezes and

binds icebergs together. At several glaciers in Greenland, the presence of a rigid mélange in front of a glacier terminus has been shown to inhibit calving and promote glacier advance, and similarly the clearing out or weakening of mélange is associated with glacier retreat (Carr et al., 2013; Cassotto et al., 2015; Fried et al., 2018; Howat et al., 2010; Joughin et al., 2008a; Kehrl et al., 2017; Kneib-Walter et al., 2021; Moon et al., 2015; Todd and Christoffersen, 2014). However, there is not always a clear relationship between mélange and terminus position (Carr et al., 2013; Sakakibara and Sugiyama, 2019).

Other work suggests that there is a relationship between seasonal terminus retreat and the timing and duration of meltwater runoff (Fried et al., 2018), or relatedly, above-freezing air temperatures (Carr et al., 2013). Runoff can drive subglacial upwelling, which increases terminus-face melting and calving due to undercutting (Wood et al., 2021). Alternatively, runoff may increase seasonal retreat via hydrofracture-induced calving (Nick et al., 2010; Sohn et al., 1998). Other studies, however, have found little or no relationship between seasonal terminus positions and runoff or its proxies such as air

temperature (Moon et al., 2015; Schild and Hamilton, 2013).

Glacier flow is highly sensitive to changes at the terminus (Howat et al., 2008; Joughin et al., 2008b; Meier and Post, 1987; Nick et al., 2009; Schoof, 2007), so short-term terminus-position variability may influence longer-term trends in glacier dynamics and, consequently, ice-sheet mass balance. Omitting terminus seasonality from numerical models can lead to both over- and under-estimated mass change projections on decadal time scales for individual glaciers, depending primarily on

the magnitude of terminus seasonality (Felikson et al., 2022). Therefore, it is important that the terminus seasonality of individual glaciers be well-characterized.

To maintain a tight focus, we limit this investigation primarily to characterizing seasonal terminus-position variability for marine-terminating outlet glaciers around the entire Greenland Ice Sheet. Given the number of processes contributing to seasonal variability (*e.g.*, surface melt, ocean temperatures, and mélange) a more detailed investigation of the causes of

seasonal variability is beyond the scope of this paper.

Our methods capitalize on the capabilities of the Sentinel-1A/B synthetic aperture radar (SAR) satellites, which typically imaged Greenland at a repeat interval of six days when both satellites were operating, and 12 days when only Sentinel-1A was operating. For the period from January 2015 through December 2021, we manually digitized monthly terminus positions for 199 glaciers around Greenland, and six- to twelve-day terminus positions for an additional 20 glaciers in central-west and





northwest Greenland. We use these terminus position data to characterize the magnitude and trends of seasonal terminus-position variability, and to estimate the frequency and seasonality of glacier retreat events at a six-day level.

## 2 Data

We used a total of 373 Sentinel-1A/B SAR mosaics (Joughin, 2020) to digitize glacier terminus positions at monthly and six or twelve-day intervals from January 2015 through December 2021. We chose this time period based on the availability of
Sentinel-1 mosaics at the time that we digitized the glacier termini (late 2021 and early 2022) and in order to capture complete years of data. The monthly glaciers are located around the full margin of the ice sheet, while the six-day glaciers are concentrated in central-west and northwest Greenland (Figure 1).

### 2.1 Satellite images

SAR mosaics of the Greenland Ice Sheet were generated from images taken by the Sentinel-1A/B satellite pair (Joughin,
2020). These satellites are able to image the ice-sheet surface regardless of cloud conditions or solar illumination, making them valuable for capturing changes in glacier behavior throughout the year. Since Version 3, the SAR mosaic product has 25 m image resolution; earlier versions had 50 m resolution. The mosaics cover 12-day intervals from 1 January 2015 through 27 September 2016, during which time only Sentinel-1A was in orbit. After the launch of Sentinel-1B, the mosaics cover six-day intervals, up until the failure of Sentinel-1B on 23 December 2021, after which the mosaics returned to 12-day
intervals. Occasionally, missed acquisitions produce intermittent spatial gaps (missing swaths) in the SAR mosaics, with corresponding temporal gaps in terminus position data for the affected glaciers.

### 2.2 Terminus positions

We manually digitized glacier terminus positions from Sentinel-1A/B SAR mosaics using ArcGIS. All digitizing was performed by a single analyst to reduce potential differences in interpretation of imagery. The error associated with these
manually-digitized terminus positions is typically comparable to the image resolution (*i.e.*, 25 m for most SAR mosaics used in this study) (Moon et al., 2015) and often results from difficult interpretation conditions arising from poor image contrast, such as extensive proglacial mélange cover.

We digitized terminus positions for a total of 219 marine-terminating outlet glaciers (Figure 1), 199 of which were digitized at monthly intervals (Table S1), and 20 of which were digitized at six-day intervals (Table S2). For the monthly glaciers, we
used the first SAR mosaic entirely within each month (*e.g.,* for the month of May, we may use 5–10 May, but not 29 April–4 May), from January 2015 through December 2021. For the 20 six-day glaciers, we chose to focus on central-west and northwest Greenland, where outlet glaciers have been changing rapidly (Black and Joughin, 2022; King et al., 2020), and selected glaciers with clear seasonal variations in the monthly data that we wanted to capture at higher temporal resolution. As part of this set, we selected all five glaciers in Upernavik Icefjord to include a local grouping of glaciers. For comparison



to those with clear seasonal variations, we included one glacier (Yngvar Nielson Gletsjer, no. 65) that did not show strong seasonal variability in the monthly data. For these six-day glaciers, we digitized terminus positions in all available SAR mosaics from January 2015 through December 2021 at 12-day intervals before October 2016 and six-day intervals thereafter. For simplicity, we refer to metrics associated with this dataset as "six-day" (*e.g.*, "six-day glaciers").

## 3 Methods

We digitized 23,333 glacier terminus positions representing January 2015 through December 2021, with an average of 82 per glacier for the monthly glaciers and 353 per glacier for the six-day glaciers. We calculated glacier area and length change and used these data to identify the significance, timing, and magnitude of terminus-position seasonality for each glacier. We also summarized these characteristics for all glaciers around the ice sheet, as well as for individual regions of the ice sheet (IMBIE, 2022; Rignot and Mouginot, 2012). Finally, we identified the timing and magnitude of individual retreat events for

the six-day glaciers.

### 3.1 Glacier area and length change

We used the box method (Moon and Joughin, 2008) to calculate glacier area over time. For each glacier, we defined an open-ended reference box with sides approximately parallel to ice flow and the back perpendicular to ice flow and upstream of the range of observed terminus positions. The box may be complex in shape (*i.e.*, composed of more than three line

segments) as it follows glacier flow around obstacles and up fjords, particularly if the glacier has retreated substantially. Each terminus position intersects both sides of the box, forming a polygon from which we calculate the area. We calculated a proxy length over time by dividing each area measurement by the mean box width at the terminus, following the methods of Black and Joughin (2022).

### 3.2 Presence of terminus-position seasonality

In characterizing terminus-position variability, one of our main objectives was to determine if there is a seasonal component to the pattern of terminus variation. To do this, we used the Lomb-Scargle periodogram, a tool for detecting periodicity in unevenly sampled data (Lomb, 1976; Scargle, 1982; VanderPlas, 2018). We chose this method due to the random temporal gaps in our time series associated with occasional missed satellite image acquisitions. For each glacier's set of terminus positions, we computed the length time series, linearly detrended the length, and calculated the Lomb-Scargle periodogram

for the detrended time series. We determined the power for cycles with a period of one year and compared this to the Lomb-Scargle false-alarm level at probability $p=0.05$. This is the threshold at which, if there were no periodic signal in the data, there could still be a peak at this frequency 5% of the time. If a glacier's length periodogram had a peak at annual frequency that exceeded the false-alarm level (*i.e.*, p<0.05), we classified it as having significant annual terminus-position seasonality.



Note that this classification only applies during our observation period (January 2015 through December 2021) as terminus-

position seasonality can change over time (*e.g.*, Joughin et al., 2008a).

### 3.3 Timing and magnitude of terminus-position seasonality

For glaciers with significant annual terminus-position seasonality (as determined with the Lomb-Scargle periodogram), we identified peaks and troughs in the glacier length time series to determine the timing and magnitude of seasonality (see example in Figure S1). First, we used a peak-finding algorithm to identify all peaks and troughs in the length data. Next, the

data were detrended and we found the resulting detrended glacier length at each peak and trough. We then found the date and detrended length of the highest peak and lowest trough for each year. In cases where retreat continued into the following year, we paired the associated trough with the peak in the previous year (*i.e.*, the peak from which the retreat initiated). We differenced the peak and trough lengths to find the magnitude of the terminus-position seasonality for each year. Finally, for each glacier, we determined the median annual dates of greatest advance and retreat, the median duration of retreat, and the

median magnitude of terminus-position seasonality. We used these values to compute the Greenland-wide and regional timing and magnitude of terminus-position seasonality.

### 3.4 Timing and magnitude of retreat events

For the six-day glaciers, we used each glacier's length time series to determine the timing and magnitude of retreat events (integrated over six days) for individual glaciers and cumulatively for the entire group of six-day glaciers. To do this, we

differenced the glacier length time series to find all potential retreat events (negative differences). To exclude small events within the range of terminus digitization uncertainty, we filtered the retreat events to retain only those with magnitudes greater than a threshold value of 50 m. In this process we do not account for glacier velocities, and so this method does not capture any retreat events that are smaller than the advance that occurred in the same time frame, *i.e.*, cases where there is net advance that is smaller than would be expected based on glacier velocity.

## 4 Results

We found that most glaciers in Greenland undergo annual cycles of advance and retreat. Figure 2 shows the relative monthly glacier length as a function of time for each of the 219 glaciers in our study. A pattern of annual cycling between relatively advanced terminus positions (blue) and relatively retreated terminus positions (red) is visible for many glaciers, illustrating terminus-position seasonality. The overall shift from blue to red over the entire duration illustrates an interannual retreat

trend. Selected length time series are shown in Figure S2 and Figure S3 to illustrate what terminus-position seasonality (or lack thereof) can look like for a subset of glaciers, including all of the six-day glaciers.





### 4.1 Prevalence of terminus-position seasonality

To better isolate the annual signals in glacier length, we computed Lomb-Scargle periodograms for the detrended data for each glacier to identify significant annual peaks. We found that between 2015 and 2021, 73.5% (*n*=161) of Greenland's marine-terminating outlet glaciers exhibited significant annual terminus-position seasonality at the 95% confidence level. For many of the other glaciers, annual peaks were visible in the Lomb-Scargle periodograms but below the 95% confidence level, suggesting some weak seasonal variability may be present. Table 1 illustrates that glaciers with pronounced seasonality are more common in western Greenland (80.9 to 94.1%) and slightly less so in the east (64.0 to 69.3%). Seasonality is least common in the north (46.2%), where several glaciers have floating ice tongues, such as Petermann (no. 93 following the MEaSUREs Glacier ID system of Joughin et al. (2015)) and Ryder (no. 96) which do not vary seasonally.

### 4.2 Timing and magnitude of terminus-position seasonality for all glaciers

We characterized the timing of terminus seasonality for the glaciers that had significant seasonality by finding the peaks and troughs in each glacier's length time series as described above. Figure 3 shows that these seasonally-varying glaciers tended to be at their most advanced state in late spring to early summer, and their most retreated state in autumn. Across all of these seasonally-varying glaciers, the median date of maximum advance ranged between May 6 and June 8, with a median of May 12. Retreat initiated immediately after the time of greatest advance, and the median date of maximum retreat ranged between September 4 and November 6, with a median of October 8. After this time, the glaciers began advancing again. The duration of the ice-sheet-wide retreat period, the time between the median dates of greatest advance and retreat, varied between 118 and 184 days, with an average of 145 days.

We calculated the amplitude of the seasonal signal in order to determine the typical annual range in detrended terminus positions. For the 161 glaciers with significant terminus seasonality, the magnitude of the terminus-position seasonality is the difference between its detrended lengths at the dates of greatest advance and retreat. Figure 4a shows a histogram of the magnitude of terminus-position seasonality. Across all of Greenland, the average annual range in terminus positions was 388 m, while the median was 221 m. Just over half of the glaciers had a magnitude of terminus-position seasonality of less than 250 m. Lower magnitudes, however, should not necessarily be interpreted as evidence of weak seasonality, as some glaciers with relatively low magnitudes showed very clear seasonal cycles. For example, Eqip Sermia (no. 5) had a magnitude of about 290 m and showed very clear terminus-position seasonality (Figure S3) (Kneib-Walter et al., 2021). Glaciers with particularly high magnitudes of terminus-position seasonality include Kangerlussuaq (no. 153; 2.65 km seasonal magnitude), Sermeq Kujalleq (Jakobshavn Isbræ, no. 3; 2.59 km), Zachariae Isstrøm (no. 107; 1.76 km), and Sverdrup (no. 46; 1.72 km). Figure 4b indicates that these glaciers with a strong seasonal variation are distributed around the ice sheet.

To look at regional variations, we organized the glaciers into six groups defined by the regional drainage basins shown in Figure 1. Table 2 shows that the timing and magnitude of terminus-position seasonality vary regionally. In the western regions of Greenland, the median dates of greatest advance and retreat tend to get later with increasing latitude. However, in



the eastern regions the dates of greatest advance and retreat occur earlier in the north than in the south. The median
magnitude of terminus-position seasonality is highest in the southwest and central-west and is lowest in the north and
northeast. The timing and magnitude of terminus-position seasonality for individual glaciers is presented in Table S3.

**4.3 Timing and magnitude of retreat events for six-day glaciers**

We digitized 20 glaciers in central-west and northwest Greenland at six-day resolution rather than monthly resolution (Table
S2). The greater temporal resolution of the six-day dataset allowed us to explore the number and magnitude of retreat events
for this subset of glaciers in northwest and central-west Greenland. Note that we use the term 'retreat events' rather than
'calving events' because our method cannot detect calving that did not offset advance between observations, and because the
calving that we did detect is integrated over a six-day period. Figure 5 shows that both the number and the magnitude of
retreat events were greatest in July and August, and were lowest in January through March. The timing and magnitude of
retreat events for these 20 glaciers, individually and combined together, are shown in Figure S4 and Figure S5.

**5 Discussion**

The data reveal several interesting points about the prevalence (Table 1), timing (Figure 3), and magnitude (Figure 4) of
terminus-position seasonality around the Greenland Ice Sheet, its regions (Table 2), and at individual glaciers (Table S3). In
the following we discuss each aspect separately.

**5.1 Prevalence of terminus-position seasonality**

Our observations show that terminus-position seasonality is widespread throughout Greenland (Figure 4b) and is especially
common in western regions of the ice sheet (Table 1). We expect that the presence or absence of terminus-position
seasonality is related to ice velocity because advection of ice is necessary for the advance phase of the seasonal terminus
position cycle. For instance, a glacier flowing at 50 m a$^{-1}$ could not sustain annual terminus-position seasonality of larger
than 50 m, because it is not flowing fast enough to replenish the ice lost each year to complete the seasonal cycle. To
estimate representative velocities, we calculated the mean velocity along the most-retreated terminus position for each
glacier, using a 2020 annual velocity map (Joughin, 2021; Joughin et al., 2010). We chose the most-retreated terminus
position to ensure that it would be covered in the velocity map. Glaciers with significant terminus-position seasonality
tended to have a much higher velocity (median velocity of 840 m a$^{-1}$) than glaciers without significant terminus-position
seasonality (median velocity of 200 m a$^{-1}$). For a glacier flowing at an average of 840 m a$^{-1}$, a seasonal retreat of 220 m (the
median magnitude of terminus-position seasonality) represents removal of a quarter of the annual advection. Applying this
relationship to the median velocity of glaciers for which we did not detect significant terminus-position seasonality, we find
that these glaciers could have a magnitude of terminus-position seasonality of about 50 m, which would be difficult to detect
in the Sentinel-1 SAR mosaics that we used.



We also explored whether our classification of the presence or absence of terminus-position seasonality aligned with other
classifications of glaciers in Greenland. Vijay et al. (2021) classified glaciers based on their seasonal velocity patterns
following Moon et al. (2014), which may indicate variations in subglacial hydrology. Their classification includes glaciers
that both speed up and slow down during the melt season ("type 2"), glaciers with high winter and spring velocities and a
longer period of slowing ("type 3"), and glaciers with no classification. We compared our terminus-position seasonality
classification with their seasonal velocity classification and found that most glaciers in most velocity categories showed
significant terminus-position seasonality. We also compared our seasonality classification with the glacier bathymetry
classification of Wood et al. (2021), who sorted glaciers into six categories based on their bathymetry at the terminus (*e.g.*,
calving on a ridge, or calving in deep fjords). Again, we found that most glaciers in most bathymetry categories showed
significant terminus-position seasonality. Thus, the lack of a clear relationship with either of these classifications suggests
that the presence or absence of terminus-position seasonality is likely not related to the type of seasonal velocity variations
or to bathymetry types. We have not established, however, whether the type of seasonal velocity variations or terminus
bathymetry have an effect on the magnitude of terminus-position seasonality.

## 5.2 Timing of terminus-position seasonality

We established that, ice sheet-wide, seasonal glacier advance tends to peak (and retreat begins) in May or early June each
year, and retreat tends to peak (and advance begins) in October to early November (Figure 3), with regional variations in
timing (Table 2). The timing of peak advance and retreat that we observe is generally consistent with regional studies of
glacier terminus seasonality (Carr et al., 2013; Fried et al., 2018; Seale et al., 2011). In cases where there are differences in
the timing of terminus-position seasonality, it appears to be related to the number of glaciers sampled (Sakakibara and
Sugiyama, 2019). The only difference in Greenland-wide compilations is with King et al. (2018), who found that ice-sheet-
wide retreat commenced about a month earlier (early April through late September) than our findings (mid-May through
early October; Table 2). However, in calculating the timing of retreat, King et al. (2018) weighted each glacier by its
contribution to total discharge; some of the highest-discharge glaciers begin retreating earlier in the year, which would bias
the weighted timing of retreat earlier in the year as well. We do find that the number and magnitude of retreat events, which
typically peaked in July and August (Figure 5), matches well with the seasonal peak discharge in mid-July reported by King
et al. (2018).

The duration of the retreat period varies from year to year (Figure 3), with 2019 and 2016 having the longest retreat periods
(180 to 184 days) and 2020, 2015, and 2017 having the shortest retreat periods (122 to 128 days); 2018 was in the middle
with a retreat duration of 156 days. We do not include 2021 here because retreat can continue into the following year, and we
have not digitized data for 2022, so our retreat duration for 2021 may be truncated. In the years with the longest retreat
periods, retreat both started earlier (early May) and ended later (early November) than in the years with shorter retreat
periods (early June through early October); in 2018 retreat occurred from early May through early October. The earlier
initiation of retreat for the years with longer retreat durations may be related to the timing of the onset of melt on the ice





sheet. We examined cumulative annual melt area (Mote, 2014; Mote and Anderson, 1995) and found that melt started relatively early (mid-April) in 2016, 2018, and 2019, and relatively late (early May) in 2015 and 2017. The 2020 result appears to be an outlier as early-season melt followed a similar trajectory to 2018 and 2019, but 2020 had the shortest

observed retreat period. The timing of the onset of melt may control the initiation and duration of retreat through the effects of increased melt on early mélange breakup, hydrofracture-induced calving, and terminus undercutting through enhanced subglacial discharge. The timing of the onset of melt appears to be more important to retreat duration than the total melt, as 2015 and 2020 ultimately had moderate cumulative melt (more than 2017 and 2018), but also had the shortest retreat durations in our record. The duration of the retreat period also does not appear to correspond strongly with annual net mass

balance or surface mass balance on an ice sheet-wide scale (Fettweis et al., 2017; Shepherd et al., 2020; Simonsen et al., 2021).

Our findings about the timing and magnitude of terminus-position seasonality provide some insights into previous studies. Fried et al. (2018) found that glaciers with terminus-position seasonality of a magnitude less than 500 m tended to be more sensitive to runoff. We found that 78% of glaciers in our study had a seasonal magnitude less than 500 m (Figure 4), so it is

possible that runoff dominates seasonality for most glaciers in Greenland. However, a number of the glaciers in our study start advancing very late in the season (*e.g.*, December, January) and/or start retreating very early in the season (*e.g.*, February, March) (Figure 3), which suggests that the timing of their seasonality is not entirely controlled by runoff. Instead, the timing of seasonality for these glaciers seems more likely to be controlled by the formation of proglacial mélange, which tends to lag the end of runoff and can facilitate glacier advance (Carr et al., 2013; Howat et al., 2010; Joughin et al., 2008a;

Kehrl et al., 2017; Kneib-Walter et al., 2021; Todd and Christoffersen, 2014), and by mid-winter episodes of mélange clearing, which can help initiate early glacier retreat (Cassotto et al., 2015; Joughin et al., 2008a). The conditions under which each mechanism may dominate remain unclear.

**5.3 Magnitude of terminus-position seasonality**

The magnitude of seasonal terminus variations tends to be small relative to the multi-decadal retreat, with 55% of glaciers

having a magnitude less than 250 m, and only 22% having a magnitude greater than 500 m (Figure 4). Many prior studies of glacier terminus seasonality used MODIS daily imagery to capture terminus positions at a higher temporal resolution than we could achieve with Sentinel-1 (Joughin et al., 2008b; Schild and Hamilton, 2013; Seale et al., 2011). However, the spatial resolution of MODIS imagery at best is limited to 250 m, so the terminus-position seasonality of many glaciers in Greenland would not be detectable in MODIS imagery. Studies using higher-resolution imagery have been focused on western

Greenland (Carr et al., 2013; Fried et al., 2018; Moon et al., 2015). Fried et al. (2018) found seasonal terminus position cycles ranging in magnitude from 150 to 1000 m in central-west Greenland, which is consistent with our findings for the same subset of glaciers, with magnitudes ranging from 80 to 880 m. The only glacier in central-west Greenland with a larger magnitude was Sermeq Kujalleq (Jakobshavn Isbræ, no. 3), with a magnitude of 2600 m, but this glacier was not included in Fried et al. (2018). In northwestern Greenland, previous estimates of magnitudes of terminus-position seasonality ranged



from 600 to 800 m (Carr et al., 2013; Moon et al., 2015), which is three to four times higher than our median magnitude of 220 m for this region. These studies looked at small subsets of glaciers in northwestern Greenland; applying approximately the same subsets to our data, we found median magnitudes ranging from 340 to 470 m and mean magnitudes between 530 and 550 m, which are still below the previously reported magnitudes. These differences in magnitude may reflect differences in methodology, as we remove the interannual length trend before calculating the magnitude of the terminus-position seasonality. Alternatively, the differences between our study and previously reported values could reflect the evolution of terminus-position seasonality over time, as the data from Carr et al. (2013) were taken from 2004 through 2012, and those from Moon et al. (2015) were from 2009 through 2014, both of which predate our study period (2015–2021).

Some previous studies have found a strong relationship between the magnitude of terminus-position seasonality and glacier width (Schild and Hamilton, 2013; Seale et al., 2011). We examined the relationship between magnitude and width for the glaciers in our study and found that, while there was a significant correlation ($p=0.004$), width alone could not explain the variance in the data ($R^2=0.051$). This correlation was improved somewhat ($p=0.000$, $R^2=0.178$) by removing Sermersuaq (Humboldt Glacier, no. 92), which at ~32 km wide was a substantial outlier (Figure S6). Seale et al. (2011) also found that glaciers with magnitudes of terminus-position seasonality of less than 1 km were typically less than 2 km wide. However, we found that of 148 glaciers with magnitudes of terminus-position seasonality less than 1 km, 107 were actually wider than 2 km.

We instead found a stronger correlation between the magnitude of terminus-position seasonality and mean annual glacier velocity ($p=0.000$, $R^2=0.479$; Figure S7). The stronger correlation between terminus-position seasonality and glacier velocity makes sense when considering how much a glacier would have to calve to balance its velocity. For instance, at a glacier that is flowing at several kilometers per year at the terminus, our median magnitude of terminus-position seasonality of 221 m would be a relatively small signal. However, above we showed that this terminus-position seasonality would remove a quarter of the annual advection for a glacier flowing at the median observed annual velocity. Because our data are detrended, the magnitude of terminus-position seasonality is separated from interannual terminus position trends. Our measured magnitude of terminus-position seasonality also only captures the period of seasonal retreat and does not include calving events that may happen outside of that period. Therefore, there is likely to be additional retreat each year to offset the annual advection and generate the widespread interannual retreat that has been observed.

**5.4 Comparisons with select individual glaciers**

The spatial breadth of our study allows us to make comparisons to data on several individual large glaciers reported in previous studies (Table 3). At all of these glaciers, we find that the differences in the timing and magnitude of terminus-position seasonality reported by our study compared to others are relatively small (approximately a month) (Howat et al., 2010; Joughin et al., 2008a, b; Kehrl et al., 2017; Schild and Hamilton, 2013). In a few cases we found larger differences in timing, typically compared to the data reported by Schild and Hamilton (2013), who studied several of these glaciers from 2001 to 2010. For example, at Daugaard-Jensen (no. 120), they reported that seasonal retreat began in late May, whereas we





found that retreat began over a month earlier, in early April. At Kangerlussuaq (no. 153), Schild and Hamilton (2013) reported that it typically retreated from July through late September; we found that it instead retreats from mid-July until

December, which is consistent with the more recent findings of Kehrl et al. (2017). Finally, at Helheim, we found that the timing of the initiation of retreat was shifted a month earlier than Schild and Hamilton (2013) and Joughin et al. (2008b), who found that Helheim typically began retreating and calving in May. All of these differences can likely be explained by interannual variations in terminus-position seasonality and the different time periods covered by these studies. Our data are insufficient to rule out potential longer-term trends in the timing of terminus-position seasonality.

**6 Conclusions**

We used Sentinel-1 SAR images to characterize terminus-position seasonality for 219 marine-terminating glaciers around Greenland from January 2015 through December 2021. We found that terminus-position seasonality is common, with 74% of glaciers expressing significant seasonality. The glaciers that do not have significant terminus-position seasonality tend to have large floating tongues or relatively low ice velocities. Of the glaciers with significant terminus-position seasonality,

retreat typically begins in mid-May and advance typically begins in early October, with some variation in different years and in different regions of Greenland, and substantial variation among individual glaciers. The number and timing of retreat events peaks in July and August and is lowest in January through March. The average annual peak-to-trough magnitude of terminus-position seasonality is nearly 400 m, although this is skewed by a few glaciers with very large seasonal cycles; the median magnitude is about 220 m. We found a stronger relationship between the magnitude of terminus-position seasonality

and glacier velocity than between magnitude and glacier width. Because glacier dynamics are sensitive to conditions at the terminus, understanding terminus-position seasonality is important for projecting future glacier change. This study provides an important step forward by extending characterizations of terminus-position seasonality from individual glaciers and regions to the entire ice sheet. The terminus positions digitized for this study may also serve as a valuable training data set for artificial intelligence-based detection of terminus positions in SAR imagery, to reduce the time and labor necessary to

produce similar data in the future.

**Code and data availability**

Data analysis and visualization code are available at https://github.com/tarynblack/greenland_terminus_seasonality. The terminus positions are being prepared for submission to NSIDC. Sentinel-1 mosaics are from MEaSUREs Greenland Image Mosaics from Sentinel-1A and -1B, Version 4 (https://nsidc.org/data/nsidc-0723/versions/4 (Joughin, 2020)).



## Author contribution

T.B. and I.J. conceptualized the project. T.B. carried out the terminus data collection, analysis, and visualization. I.J. prepared the SAR data products. T.B. prepared the manuscript, with contributions from I.J.

## Competing interests

The authors declare that they have no conflict of interest.

## Acknowledgements

T.B. and I.J. were supported by the NASA MEaSUREs program (80NSSC18M0078). T.B. thanks Michalea King and Twila Moon for conversations that improved the manuscript.

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





**Figures**

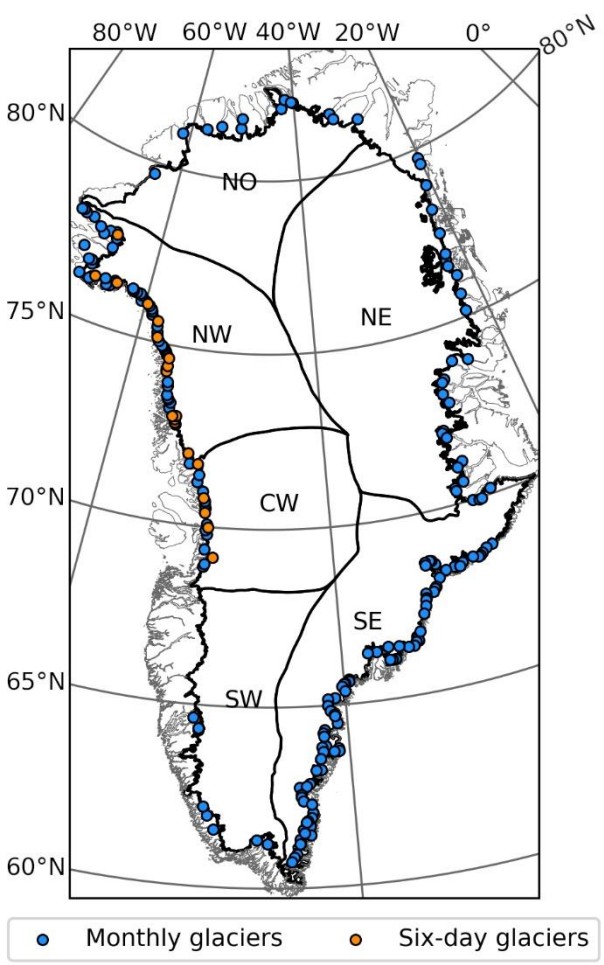

**Figure 1. Map of glaciers covered in this study. Glaciers with termini digitized at monthly resolution are in blue, and glaciers with termini digitized at six-day resolution are in orange. The regions outlined are southwest (SW), central-west (CW), northwest (NW), north (NO), northeast (NE), and southeast (SE) Greenland.**

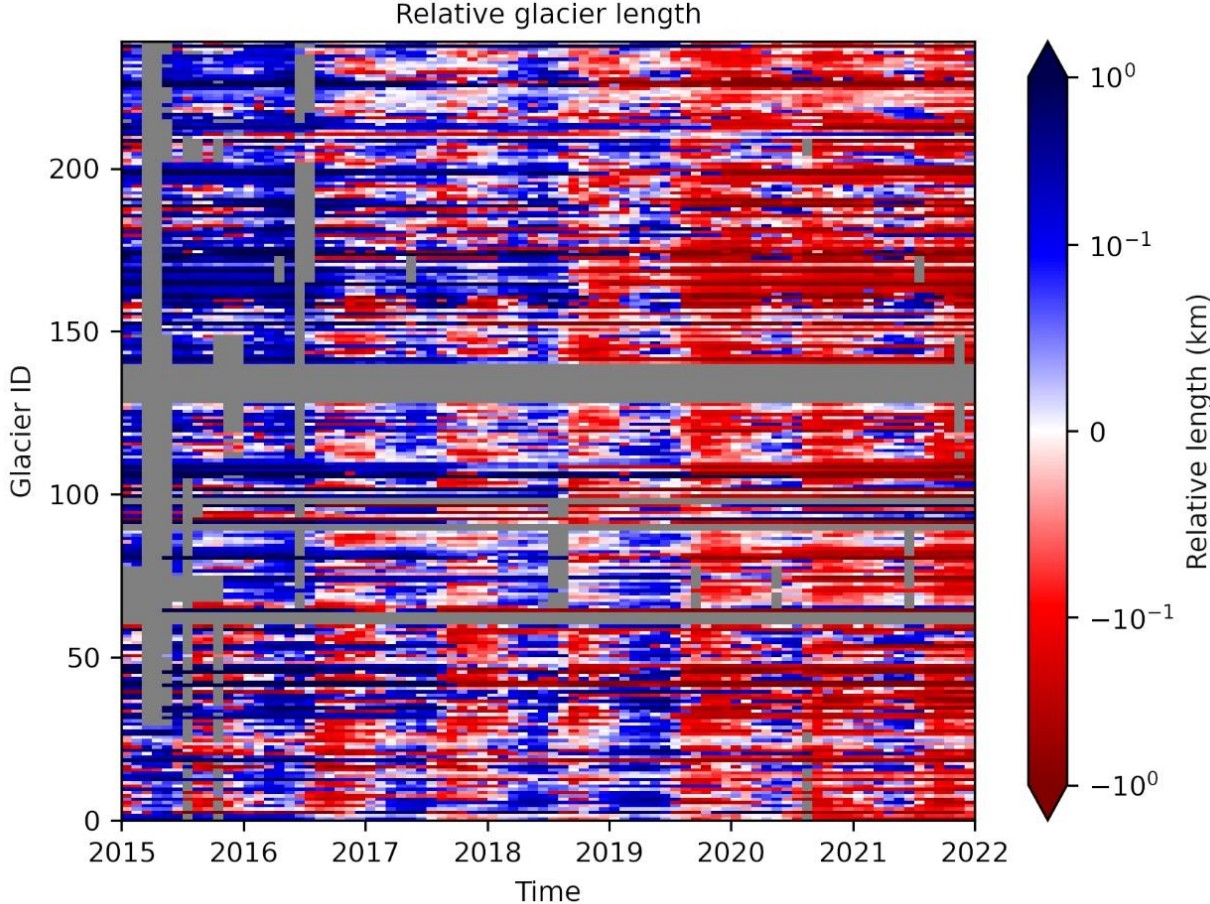

**Figure 2. Glacier length relative to mean 2015–2021 length for each glacier. The Glacier ID system is derived from MEaSUREs glacier data (Joughin et al., 2015) and is detailed in Table S1 and Table S2. The color scale is logarithmic with a linear threshold at ±10⁻¹ to account for values approaching zero. Terminus positions that are advanced relative to the mean length appear in blue, and terminus positions that are retreated relative to the mean length appear in red. No-data values are gray and are due to either spatial or temporal gaps in the SAR mosaics used for digitizing terminus positions.**


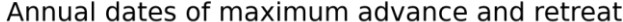

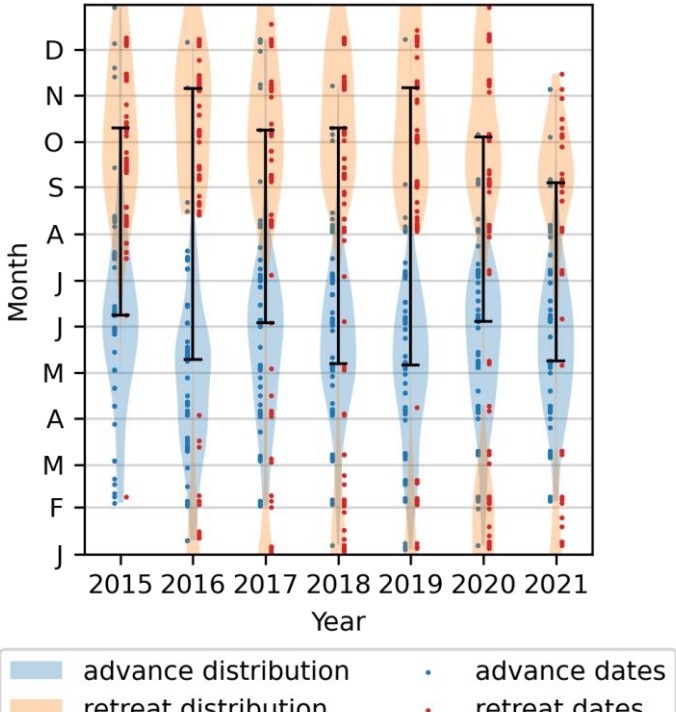

**Figure 3. Distribution of the annual timing of maximum terminus position advance (blue) and maximum terminus position retreat (orange) for all seasonally-varying glaciers in the study. The width of the shaded region represents the likelihood that maximum advance (blue) or retreat (orange) will occur on a given date. The vertical black bars show the duration of the retreat period each year, and the horizontal lines at the end of the vertical bars show the median date of maximum advance (in the blue region) or retreat (in the orange region) for each year. Dots show the dates of maximum advance (in blue, to the left) or maximum retreat (in orange, to the right) for individual glaciers each year. Retreat timing distributions in the early months of a year are continuations of the retreat timing distributions from the previous year.**





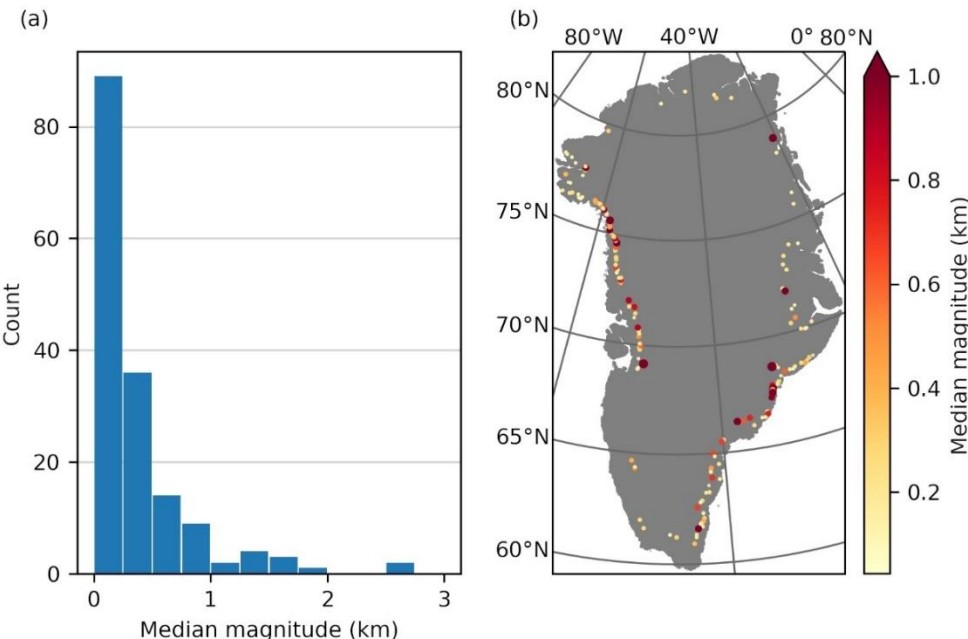

**Figure 4. (a) Histogram and (b) map of the distribution of the median magnitude of terminus-position seasonality for all seasonally-varying glaciers in the study. In (b) both the color and size of the points represent the magnitude of terminus-position seasonality.**

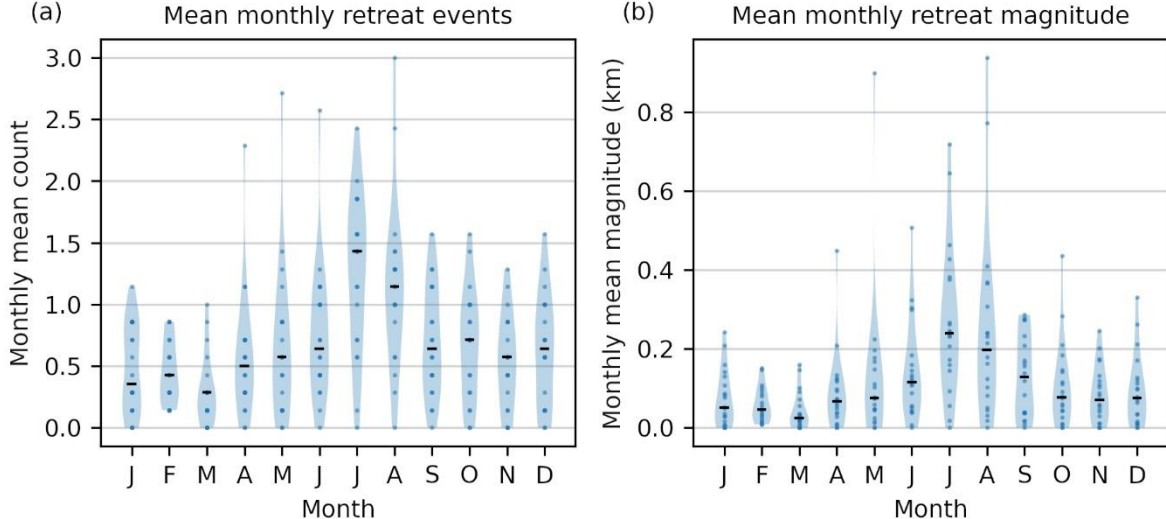

**Figure 5. Distribution of the monthly (a) number and (b) magnitude of retreat events for each of the six-day glaciers. The width of the shaded region represents the likelihood of a given (a) number or (b) magnitude of retreat events occurring in a given month. Horizontal black bars show the median for each month, and blue dots show the values for individual glaciers.**





**Tables**

**Table 1. Regional breakdown of the number and percentage of glaciers with significant (p<0.05) terminus-position seasonality.**

| Region | Significant glaciers | Total glaciers | Percent significant |
|---|---|---|---|
| Southwest (SW) | 7 | 8 | 87.5% |
| Central west (CW) | 16 | 17 | 94.1% |
| Northwest (NW) | 55 | 68 | 80.9% |
| North (NO) | 6 | 13 | 46.2% |
| Northeast (NE) | 16 | 25 | 64.0% |
| Southeast (SE) | 61 | 88 | 69.3% |
| **Greenland** | **161** | **219** | **73.5%** |

500 **Table 2. Regional breakdown of the timing and magnitude of terminus-position seasonality. All reported values are medians across the set of glaciers in a given region. To contextualize the magnitudes, note that the terminus digitization uncertainty is typically ~25 m.**

| Region | Peak advance | Peak retreat | Retreat duration (days) | Magnitude (m) |
|---|---|---|---|---|
| Southwest (SW) | 20 April | 7 September | 156 | 273 |
| Central west (CW) | 1 May | 9 September | 153 | 225 |
| Northwest (NW) | 20 May | 7 October | 143 | 221 |
| North (NO) | 4 June | 14 September | 108 | 203 |
| Northeast (NE) | 6 May | 5 October | 148 | 201 |
| Southeast (SE) | 25 May | 13 November | 177 | 221 |
| **Greenland** | **12 May** | **8 October** | **154** | **221** |

505 **Table 3. Timing and magnitude of terminus-position seasonality for several individual glaciers for comparison with previous studies. All reported values are medians for the given glacier.**

| Glacier | Peak advance | Peak retreat | Retreat duration (days) | Magnitude (m) |
|---|---|---|---|---|
| Sermeq Kujalleq (Jakobshavn) | 16 April | 14 August | 120 | 2590 |
| Kangilliup Sermia (Rink) | 4 June | 22 September | 110 | 810 |
| Daugaard-Jensen | 8 April | 2 September | 147 | 1330 |
| Kangerlussuaq | 2 July | 26 December | 177 | 2650 |
| Helheim | 7 April | 24 September | 170 | 1480 |