# Peer review of "Weekly to monthly terminus variability of Greenland's marine-terminating outlet glaciers"

_The Cryosphere, 2022_

## Author Comment (AC1)

**Response to Reviewer #1 on "Weekly to monthly terminus variability of Greenland's marine-terminating outlet glaciers"**

Comment received: 7 October 2022

Key:
Reviewer comment (blue)
Response (black)

Summary:
This paper presents a Greenland-wide dataset of marine-terminating outlet glacier terminus positions. Front position timeseries for almost 200 glaciers are presented at monthly resolution between 2015 and 2021, with an additional twenty glaciers recorded at 6-day resolution. The paper focuses on simply characterising the timing and magnitude of seasonal terminus position change. The authors find that about 75% of their studied glaciers display significant seasonal variability in front position. Such seasonality had already been observed at roughly monthly frequency at many of Greenland's glaciers, but existing studies generally treated smaller samples of glaciers. The 6-day resolution data represent a more substantial advance, with similar temporal frequency observations existing only for a few of the largest glaciers.
The paper is very clearly and succinctly written and the dataset seems robust, is on the whole explained thoroughly, and represents an enormous amount of painstaking work which will prove useful to the glaciological community. The data analysis and comparison with other existing datasets, however, misses several opportunities for more thorough investigation, especially for the 6-day data, resulting in somewhat insubstantial conclusions.

We thank the reviewer for their comments, which have helped to clarify the writing and improve the analysis.

Main points:
1. Identification of glaciers with prominent annual front position variability: How precise was the requirement for an annual frequency in the periodogram? Would a frequency of 380 days count, or 400 days? It might be worth bracketing around a precise annual frequency (e.g. by one month either side for the monthly data, in smaller increments for the 6-day data) to account for the effect of shorter-term variability superimposed on the annual signal. Perhaps this is already done, but if so, the process is not fully explained in the manuscript.

We were initially only comparing the power to the 95% confidence level at exactly 1.0 yr$^{-1}$. However, after examining the Lomb-Scargle power for a sinusoid with a period of one year, we broadened our analysis to include powers between 0.9 and 1.1 yr$^{-1}$ (the width of the power peak for the sinusoid). That is, if a glacier's length data has a peak in power anywhere between 0.9 and 1.1 yr$^{-1}$ that exceeds the 95% confidence level, the glacier is classified as having significant seasonality. We have updated the text throughout section 3.2 to reflect this change in methodology. Due to this change, an additional ten glaciers were classified as seasonal, which resulted in minor changes to various numbers throughout the text, but no changes in overall interpretation.

2. Unambitious analysis and incomplete comparison with existing datasets: The data analysis and comparison with other existing datasets, however, misses several opportunities for more thorough investigation. For example, the authors could compare the timing and magnitude of seasonal front position variability in their 6-day data with glaciological and climatological

factors (rather than simply referring to agreement or otherwise with glacier groups from other datasets). In addition, the comparison with existing terminus position datasets conflates differences in terminus position change that could be due to either varying methodologies and source data, different study periods or a real change in seasonality through time. As the authors admit, they can't differentiate these potential causes. It would be much more useful to find an existing dataset which overlaps with their own but also extends further back in time (e.g. PROMICE, ESA CCI?). Then they could try to quantify any real longer term temporal changes.

We have already compared longer-term terminus data (a multidecadal dataset that includes some of the data presented in this manuscript) with several other datasets (MAR, ECCO, *etc.*) to explore the effects of climatological factors at annual resolution (Black and Joughin, 2022; doi:10.5194/tc-16-807-2022). There were few datasets that overlapped both spatially and temporally (including more recent years) with our data in that study, and we did what we could to compare and identify any major differences. In that study we also found it challenging to identify distinct causes of terminus retreat with the data available, essentially because each type of data could represent multiple mechanisms of retreat. This analysis would be even more difficult to accomplish with the data in this study, and we decided that such an analysis is beyond the scope of this manuscript.

Line by line points:

L7: This does not seem like the most effective way to start the abstract (talking about earlier studies). Perhaps it would be better to say something like: 'Seasonal terminus-position variability of Greenland's marine-terminating outlet glaciers is superimposed on multi-decadal trends of…'
We have changed this sentence as suggested.

L57-60: It is somewhat disappointing that the paper does not pursue potential causal factors of the observed seasonal variability, especially for the 6-day data, which represent a good opportunity for such exploration at a broader range of outlet glaciers than previous studies.
Exploring the causal factors in detail would require extensive additional analysis on the scale of another paper (*e.g.*, we performed a similar analysis for multidecadal data in Black and Joughin, 2022, doi:10.5194/tc-16-807-2022). This would be a good subject for future research. On the suggestion of Reviewer 2, we did edit the last sentence of those mentioned here to clarify that we do briefly discuss potential causal factors, as follows:
"…a more detailed investigation of the causes of seasonal variability is beyond the scope of this paper; rather, we discuss the potential role these factors may have."

L76: When did version 3 come online? What proportion of the images used had a resolution of 50 m and what proportion 25 m? Did the authors undertake a comparison on a date with both image resolutions to quantify any potential impacts of the change in image resolution on the consistency of their front position timeseries?
We have changed the text to read:
"Since Version 3, which was released in August 2020, the SAR mosaic product has 25 m image resolution; the majority (85.8%) of termini in this study were digitized using the higher-resolution mosaics. Earlier versions of the SAR mosaic product had 50 m resolution, and the remaining 14.2% of termini were digitized using these lower-resolution products as part of an earlier study."

We did not perform a comparison of terminus digitizations at different image resolutions. The lower-resolution termini were digitized as part of an earlier study with mixed-resolution (15-50m) images and we did not encounter any issues with the different resolutions. The lower-resolution termini also represent a comparatively small portion of all of the termini in this study.

L120: How strict was the adherence to a 'one year' frequency? (see Main point 1).
Per our response to the Main Point 1, we updated the methodology to search for power peaks between 0.9 and 1.1 $yr^{-1}$ frequencies rather than only at 1.0 $yr^{-1}$.

L124-125: Could the seasonality not also have changed during your 6-year study period? Perhaps also employing a Lomb-Scargle wavelet scalogram might be able to detect temporal changes in the dominant frequency?
We are already trying to pull a dominant frequency out of only seven cycles, and the seasonality can clearly change from year to year, so we are not going to be able to detect significant trends in seasonality over such a short time frame. However, it is good to more clearly identify the interannual variability in seasonality, and we have added this in section 4.2 as follows:
"The annual median magnitude of ice-sheet-wide terminus-position seasonality ranged from 200 to 275 m (Table 2)."
The Table 2 that is referenced is a new table which outlines the median magnitude of terminus-position seasonality and the duration of the retreat period for each year during the study period.

L129-132: I wonder if it might be better to initially smooth the data before picking out the peaks and troughs? Otherwise you risk biasing the underlying frequency by incorporation transient changes in front position (e.g. Daugard-Jensen Glacier 2017 & 2018, Figure S3). This may also change the number of glaciers with significant 'annual' periodicity in their front position records.
This comment is unclear to us as the peak-finding process for glacier lengths is completely separate from the analysis used to identify glaciers with significant periodicity. Perhaps our wording was unclear in this section. We have rewritten this part of section 3.3. to better distinguish between these analyses as follows:
"If a glacier was determined to have significant annual terminus-position seasonality, we then used a peak-finding algorithm to identify all peaks and troughs in the original length data. Next, the length data were linearly detrended and we found the resulting detrended glacier length at each peak and trough. We then found the date and detrended length of the highest peak and lowest trough for each year. In cases where retreat continued into the following year, we paired the associated trough with the peak in the previous year (i.e., the peak from which the retreat initiated)."

L136: It might be worth reiterating here that the timing (for this part of your dataset) can only be determined to monthly resolution.
We added "at a monthly resolution" to the end of the sentence.

L225-226: This would seem like a relatively straightforward and worthwhile thing to do (you have the data, especially for the 6-day glaciers). Could the authors justify their reasoning for omitting this avenue of further investigation?
We have reorganized the final paragraph of section 5.1 and added the following analysis, including two new supplementary tables (Tables S4 and S5):

"We also explored whether our classification of the presence or absence of terminus-position seasonality aligned with other classifications of glaciers in Greenland. Vijay et al. (2021) classified glaciers based on their seasonal velocity patterns following Moon et al. (2014), which may indicate variations in subglacial hydrology. Their classification includes glaciers that both speed up and slow down during the melt season ("type 2"), glaciers with high winter and spring velocities and a longer period of slowing ("type 3"), and glaciers with no classification. We compared our terminus-position seasonality classification with their seasonal velocity classification and found that most glaciers in most velocity classes showed significant terminus-position seasonality (Table S4), which suggests that the presence or absence of terminus-position seasonality is likely not related to the type of seasonal velocity variations. There may, however, be some correspondence between the type of seasonal velocity variations or terminus bathymetry and the magnitude of terminus-position seasonality. The Vijay classification leads to clusters defined by both seasonal magnitude and average speed: type 2 glaciers tend to have lower magnitudes and slower flow than type 3 glaciers. This relationship between seasonal magnitude and speed is not surprising because, as described above, larger seasonal retreats are required to balance greater velocities.

We also compared our seasonality classification with the glacier bathymetry classification of Wood et al. (2021), who sorted glaciers into six categories based on their bathymetry at the terminus (*e.g.*, calving on a ridge, or calving in deep fjords). Again, we found that most glaciers in most bathymetry categories showed significant terminus-position seasonality (Table S5), which suggests that the presence or absence of terminus-position seasonality is probably unrelated to the type of bathymetry. While in general deeper bathymetries should correspond to faster speeds, the Wood classification factors in both bed shape and depth; consequently, the classifications of deeper glaciers and glaciers calving on a ridge likely span an overlapping range of depths. The results do suggest, however, that glaciers with a stabilizing ridge, despite their generally faster speeds, undergo less seasonal variation than do deep-water glaciers with no ridge. The glaciers classified as shallow (<100 m) are some of the glaciers with the slowest speeds and the least retreat, consistent with the correspondence between velocity and retreat that we observe."

L318: Given the different periods covered by the previous studies and the data generated by the present study, I wonder about the value of these comparisons as presented. I think a more detailed comparison would be valuable, however. (see Main point 2).

A detailed comparison with earlier datasets would be a great topic for future study, but in this manuscript we chose to focus on presenting our dataset and the characteristics of terminus-position seasonality derived from our data. We included this section (5.4) to place our work in the context of what has already been done.

---

## Author Comment (AC2)

**Response to Reviewer #2 on "Weekly to monthly terminus variability of Greenland's marine-terminating outlet glaciers"**

Comment received: 24 October 2022

Key:
Reviewer comment (blue)
Response (black)

Summary:
In the manuscript Black and Joughin present a monthly (to sub-monthly) dataset of terminus positions for >200 marine-terminating glaciers of the Greenland Ice sheet between 2015 and 2021. The authors use this data set to characterise and evaluate seasonal variability. Specifically, they find: 75 % of their studied glaciers show significant seasonal variability; seasonal retreat to begin in mid-May and advance in early October, on average; retreat events to peak in late summer and reach a minima in late winter; and a stronger correlation between seasonal magnitude and velocity as opposed to magnitude and glacier width.
Whilst several aspects of the work have been reported on before, the scale at which the analysis has been conducted is unique and provides new insights into ice-sheet-wide trends of seasonal terminus position variability and—for the six-day data—the frequency and seasonality of glacier retreat events.
Overall the manuscript is well written and structured, and the dataset will be a useful (and impressive) addition to the growing availability of ice-sheet-wide studies of glacier termini. I recommend the manuscript be accepted after some minor revisions outlined below.

We thank the reviewer for their comments, which have helped to clarify the writing and improve the analysis.

Main points:

1. As highlighted in the review of RC1 (07 Oct 2022), the data analysis and comparison with other data (e.g. Section 5.3 and 5.4) misses opportunity for more thorough investigation. For example, furthering their comment 2, there is no real investigation/discussion on how or if seasonality has changed over your study period, or longer, at glacier or sector scale. This is important as highlighted in the abstract and elsewhere (e.g. Felikson et al. [2022] doi:10.1029/2021JF006249) seasonal fluctuations can influence longer-term glacier dynamics. A more thorough analysis here would certainly strengthen the manuscript and further highlight the glaciological application of the dataset.

Given the short duration of our study period and the high interannual variability, we are not going to be able to detect significant trends in seasonality over our study period. However, it is valuable to characterize the interannual variability in the timing and magnitude of seasonality, and we have done so in a new Table 2 which outlines the duration and magnitude of the retreat period for each year during the study period. In time, with more data, this would be an excellent avenue of research. For longer-term data at annual (not seasonal) temporal resolution, we refer to our recent publication on glacier retreat and climatic drivers in northwestern Greenland (Black and Joughin, 2022; doi:10.5194/tc-16-807-2022).

2. One of the more novel findings of the work is the association between seasonal magnitude and velocity as opposed to glacier width, but some of this analysis is buried in the supporting material. It would be worth merging Figures S6 b and S7 and placing these in the main manuscript.

Great suggestion – we have merged Figures S6b and S7 to create a new Figure 6 in the main manuscript. We have removed Figure S6a entirely.

Specific points:
1. Lines 57 -60. You specify that a detailed investigation of the causes of seasonal variability is beyond the scope of the manuscript, but include some basic analysis and arguments in this regard in Section 5.2. Consider rewording to better highlight the comparisons. '…beyond the scope of the paper, rather we discuss the potential role these factors may have.'?

We have edited the last sentence as follows:

"…a more detailed investigation of the causes of seasonal variability is beyond the scope of this paper; rather, we discuss the potential role these factors may have."

2. Lines 80-81. Please provide an indication of how much data is missed (e.g. median % coverage of the glaciers).

We added the line:

"On average, over the seven-year study period, these missed acquisitions resulted in two missing data points for monthly glaciers, and 20 missing data points for six-day glaciers (accounting for the transition from 12-day to six-day acquisitions)."

3. Lines 107-113. This method feels more akin to the curvilinear box method. If so I'd also reference Lea et al. (2014) doi: 10.3189/2014JoG13J061.

We modified a sentence in this paragraph to read:

"…particularly if the glacier has retreated substantially; in this way it is comparable to the curvilinear box method (Lea et al., 2014)."

We do wish to note that the boxes we used are the original boxes of Moon and Joughin (2008), with modifications as needed if glaciers have retreated past the original boxes, so our citation of that method was quite literal. The Moon and Joughin boxes were never the narrow three-sided ones (which often don't capture much of the front) that are seen in some other studies; we are trying to correct this common misconception.

4. Line 130. How were the data detrended? Linear as per Section 3.2?

Yes, linearly; we restated as "linearly detrended".

5. Line 142. 'threshold value of 50 m' à 'threshold value of 50 m (i.e. 2σ)'

Changed as suggested.

6. Line 163. '...as described above' à '…as described in Section x.x'.

Changed as suggested (Section 3.3).

7. Line 240-242. Have you explored if there is any relationship between duration of retreat and magnitude of retreat? Could be worth exploring?

We added a new Table 2 which shows the annual ice-sheet-wide magnitude and duration of retreat. In section 4.2 we added that "The annual median magnitude of ice-sheet-wide terminus-position seasonality ranged from 200 to 275 m (Table 2)." We edited the lines noted in this comment (in section 5.2) to read:

"The duration of the retreat period varies from year to year (Figure 3). The annual magnitude of ice-sheet-wide terminus-position seasonality tends to increase with the duration of the annual retreat period (Table 2); although the sample size is small, linear regression indicates a strong fit ($R^2$=0.803, $p$=0.016) between magnitude and duration. The years 2019 and 2016 have the longest retreat periods…"

8. Line 266. Add references to Howat et al. (2010) doi: 10.3189/002214310793146232 and Bevan et al. (2019) doi: 10.5194/tc-13-2303-2019.

We have added the suggested references.

9. Line 315. More recent papers by Brough et al. (2019) doi: 10.3389/feart.2019.00123 and Bevan et al. (2019) doi: 10.5194/tc-13-2303-2019 highlight similar findings for Kangerlussuaq and cover more of your study period. Include these references here too.

We have added the suggested references.